# Emissions Reduction Target Plan and Export Product Quality: Evidence from China's 11th Five-Year Plan

**Xingye Zhou and Helian Xu \***

School of Economics and Trade, Hunan University, Changsha 410006, China; zhoudoint@163.com
\* Correspondence: xuhelian@hnu.edu.cn; Tel.: +86-13037319676

**Abstract:** Determining how environmental policy affects export competitiveness is essential for achieving win–win objectives in environmental governance and trade development. We examine whether and how China's emissions reduction policy declines the firm's export product quality during the 11th five-year plan via the difference-in-difference-in-difference method. The main findings of this paper are as follows: (i) Compared with less polluting industries, the export production quality declines 5.74% and 3.31%, respectively, as the pollution reduction targets of $SO_2$ and chemical oxygen demand (COD) are set 1 unit higher in more polluting industries. (ii) The negative effect is greater in Western regions as well as state-owned firms. (iii) Local officials facing promotion pressure are more incentivized to raise emissions reduction goals, as well as greater increasing emissions abatement costs, thus leading to greater declines in export quality. However, the innovation compensation effect still remains. (iv) The negative effects can be mitigated through product switching that contributes to resource allocation within firms towards their core products, or executing the first-mover advantage in response to the emissions reduction plan.

**Keywords:** emissions reduction target plan; export product quality; core products; first-mover advantage

## 1. Introduction

After more than 40 years of reform and opening-up, China has created an export miracle relying on its immense demographic dividend as well as cheap resources. However, in addition to integrating into the global value chain with low-value-added products, China has also triggered a growing number of environmental problems. Thus, China's manufacturing sectors face a double whammy, internal environmental constraints and external green barriers to exports, and the upgrade to export quality is expected soon. In 2006, the State Council issued the "11th Five-Year Energy Conservation and Emission Reduction Work Plan", which proposed major pollutant mitigation targets, and linked them to local officials' promotions. This policy not only influences firms' export scales but also, more importantly, export quality.

The relationship between environmental policy and export competitiveness has gained considerable attention in recent years. The starting point of this investigation is Porter's hypothesis that a well-designed environmental policy can trigger innovation which can offset the compliance costs of environmental regulation [1,2]. Subsequent discussion of the Porter hypothesis, such as Martín-Tapia et al. [3], Costantini and Mazzanti [4], Elrod and Malik [5], Joo et al. [6], and Chen et al. [7], has supported the view that environmental policy promotes exports.

However, scholars have argued that stringent environmental policies weaken firms' export competitiveness [8,9]. This is largely because compliance cost effects caused by environmental policy not only increase firms' production costs, but also decrease the inputs of production factors [8]. Thus, many countries distort levels of environmental regulations as an important pattern of protecting domestic industries [10,11]. Moreover, strict environmental policy decreases both export likelihood and export values by preventing new entries to the export markets, as well as reducing the amount of exporting destinations [9,12].

It should be noted that most of the above analyses of export competitiveness are based on traditional indexes such as export volumes and shares. These indicators reflect the export scales of relevant economies. The export scale advantage which accompanies intensifying vertical specialization does not necessarily mean improvement in the international division of labor [13,14]. New trade theory posits that product quality is a new advantage of export competitiveness [15,16]. Therefore, understanding the relationship between product quality and environmental regulations is important for environmental protection and policy making [17,18].

Regarding China's 11th Five-Year Plan as a quasi-natural experiment, we construct a difference-in-difference-in-difference (DDD) model to analyze the effect of its emissions reduction plan on export product quality. We begin by comparing export product quality for firms located in provinces with higher emissions reduction targets to those with lower emissions reduction targets, before and after the 11th Five-Year Plan. Then, we compare these estimates for firms in polluting or dirty industries. The main contributions of this study are as follows:

Firstly, whereas previous studies have used the 11th Five-Year Plan as a quasi-experiment to investigate the economic effects of environmental regulation [9,12,19,20], we are the first to introduce double emissions reduction targets into our framework and consider how $SO_2$ and chemical oxygen demand (COD) emissions plans affect firms' export performance by comparing the before-and-after changes among firms in industries with different emissions reduction targets in different provinces.

Secondly, we examine the possible mechanisms through which the emissions reduction target policy may lead to a decrease in export quality. Not only do we examine in terms of firms' response, but we also highlight the institutional mechanisms through which political pressure motivates local officials to adopt various trade-off strategies to improve export quality to reach more stringent emissions reduction targets. In this way, we add to the literature on what motivates local governments to adopt environmental regulation [21,22].

Finally, we investigate how firms may overcome emissions reduction plans' negative effects on export product quality. Existing studies have emphasized the role of product category switching in resource allocation [16,23–25]. In this paper, we investigate the extent to which the distance to a core product can mitigate the negative effects of a new environmental policy. Concerning firms without product switching, we further illustrate how to capitalize on the first-mover advantage or past experience to attenuate any negative effects of emissions reduction target policy.

The remainder of this paper is organized as follows. Section 2 details the policy background of the emissions reduction target policy. Section 3 introduces the models and describes the data issues. Section 4 reports the main modelling results. Section 5 extends the baseline model to address potential mechanisms, as well as countermeasures to mitigate any negative effects introduced by the new environmental policy. Section 6 concludes and outlines our study's policy implications.

## 2. Policy Background of the Emissions Reduction Target Policy

Since establishing its system of environmental regulation in the late 1970s, China has been committed to curbing environmental pollution. In 1987, the Chinese government enacted The Atmospheric Pollution Prevention and Control Law, as a measure to improve air quality. Then, in 1998, the State Council implemented the Two Control Zone (TCZ for short) policy, and 175 cities were designated TCZ cities [26]. After that, the 10th Five-Year Plan (2001–2005) set a total $SO_2$ emissions reduction target of 10% of the national level at the time. However, no provincial reduction targets were set, and thus, implementation was poor [12].

To further curb environmental pollution, more binding policies to control pollution discharges were proposed in the 11th Five-Year Plan. This new pollution-mitigation policy led to the implementation of a variety of environmental regulations in all regions of China (rather than the TCZ areas only). The proposed environmental regulations were more

prominent, highlighted by the following two components. Firstly, the State Council set a long-term emissions reduction goal for each provincial and local government, and the Central government set a total reduction target of 10% of major emissions at the national level and announced a mandatory target responsibility system to curb pollution discharges.

Subsequently, authorized by the State Council, the National Environmental Protection Agency signed a contract with the local governments of the 31 provinces, municipalities, and autonomous regions. The contract stipulated each province's emissions reduction targets for $SO_2$ and COD (Table 1). The allocation principle behind the major emissions reduction targets was holistically considering provincial differences in environmental quality, environmental capacity, emissions amounts, and economic growth on the premise that the national reduction target was attainable. Another principle was that the Eastern, Central, and Western regions would be treated differently and would be required to implement differentiated reduction targets. In addition, the work plan reported that the total target amount of $SO_2$ emissions was 22,944 thousand tons, of which 22,467 thousand tons was allocated to local governments and 0.477 thousand tons was reserved for an emissions trading pilot program. The total COD emissions target amount was 127.28 million tons, of which 126.39 million tons was allocated to local governments and 8900 tons was reserved. Thus, the new emissions-reduction plan was characterized by a top-down decomposition approach which was stricter than previous environmental strategies which had been characterized by overall quantity control.

**Table 1.** Pollution reduction targets (%) by province and region.

| Eastern | $SO_2$ | COD | Central | $SO_2$ | COD | Western | $SO_2$ | COD |
|---|---|---|---|---|---|---|---|---|
| Beijing | 20.4 | 14.7 | Shanxi | 14 | 13.2 | Inner Mongolia | 3.8 | 6.7 |
| Tianjin | 9.4 | 9.6 | Jilin | 4.7 | 10.3 | Guangxi | 9.9 | 12.1 |
| Hebei | 15 | 15.1 | Heilongjiang | 2 | 10.3 | Chongqing | 11.9 | 11.2 |
| Liaoning | 12 | 12.9 | Anhui | 4 | 14.9 | Sichuan | 11.9 | 5 |
| Shanghai | 25.9 | 14.8 | Jiangxi | 7 | 5 | Guizhou | 15 | 7.1 |
| Jiangsu | 18 | 15.1 | Henan | 14 | 10.8 | Yunnan | 4 | 4.9 |
| Zhejiang | 15 | 15.1 | Hubei | 7.8 | 5 | Tibet | 0 | 0 |
| Fujian | 8 | 4.8 | Hunan | 9 | 10.1 | Shaanxi | 12 | 10 |
| Shandong | 20 | 14.9 | | | | Gansu | 0 | 7.7 |
| Guangdong | 15 | 15 | | | | Qinghai | 0 | 0 |
| Hainan | 0 | 0 | | | | Ningxia | 9.3 | 14.7 |
| | | | | | | Xinjiang | 0 | 0 |
| Average | 14.4 | 12 | Average | 7.8 | 10 | Average | 6.5 | 6.6 |

Note: $SO_2$ and COD reduction targets are collected from the document named "Objectives and Responsibilities in Reducing the Total Amount of Major Pollutants During the 11th Five-Year Plan" issued by the China State Council in 2006.

The second component of the environmental regulations was that the Central government linked the major pollutant reduction targets to local officials' promotions. The National Development and Reform Commission issued a document entitled "Decision on Implementing Scientific Development Concepts and Strengthening Environmental Protection" on 3 December 2005. It declared environmental protection an important criterion for cadre assessment and official promotion. This promoted a shift in local officials' assessment criteria from an economic performance index to an environmental–economic indicator. Environmental performance became an essential component of cadre selection and appointment. This marked the first time the Central government enforced the environmental protection "accountability system" and the "one-veto negation system", making environmental performance an essential component of cadre selection and appointment. The State Council then promulgated "the Comprehensive Work Plan for Energy Conservation and Emissions Reduction". Additionally, the National Environmental Protection Agency, the National Statistics Bureau, and the National Development and Reform Commission were required to disclose discharge data every 6 months and conduct annual inspections and

assessments beginning in 2006. In 2008, an interim assessment of the results of the pollution reduction targets was conducted, and the final assessments were conducted in 2010.

Statistics showed that 27 provinces had established performance appraisal in environmental management and had incorporated environmental protection into their assessment systems for economic and social development. Moreover, 21 provinces had incorporated environmental protection into their assessment systems for cadre achievements, first establishing a chief cadre responsibility system in environmental protection. Emissions reduction goals had produced impressive results. Total $SO_2$ and COD were decreased by 14.4% and 12.5%, respectively, during the 11th Five-Year Plan, and all provinces reached their emissions reduction targets.

## 3. Empirical Specifications and Data

### 3.1. Model Specifications

Combining the variation in the emissions reduction targets across provinces and export product quality data from before and after the 11th Five-Year Plan, we conducted a quasi-difference-in-difference model to capture how emissions reduction policy affects firms' export behavior.

One concern about quasi-difference-in-difference analysis is that some time-varying regional characteristics simultaneously correlate with both the outcome variables and the regressor, thus biasing our estimates [12,26]. In light of this concern, we exploited the fact that industries which have different pollution emissions are affected by emissions reduction policy differently, and we conducted a difference-in-difference-in-difference (DDD) estimation to control for industry differences. Our specification was as follows:

$$
\begin{aligned}
lnquality_{fict} \; &= \beta_0 + \beta_1 lntarget_{1p} \times Post_t \times lnSO_{2i} + \beta_2 lntarget_{2p} \times Post_t \times lnCOD_i + \mu_{pt} + \gamma_{ht} + \delta_{pi} \\
&+ \varphi_{fc} + \varepsilon_{fict}
\end{aligned}
\tag{1}
$$

where $Quality_{fict}$ is the export quality for a product in industry $i$ exported by firm $f$ to destination country $c$ in year $t$, and $Target_p$ represents the pollution reduction target for province $p$ in which firm $f$ is located. This can be further divided into $SO_2$ reduction targets ($target1$) and COD reduction targets ($target2$). $Post_t$ is a dummy variable which equals 1 for 2006–2010, or 0 otherwise, $SO_{2i}$ denotes the $SO_2$ emissions intensity in industry $i$, while $COD_i$ denotes the COD emissions intensity in industry $i$.

Province–year fixed effects ($\mu_{pt}$), industry–year fixed effects ($\gamma_{ht}$), province–industry fixed effects ($\delta_{pj}$), and firm–destination fixed effects ($\varphi_{fc}$) were controlled. To address potential heteroskedasticity and serial correlation, we clustered the standard errors at the province–industry level.

### 3.2. Data Source

We constructed data samples from three official statistical databases. The pollution reduction target data were from "The Documents of Objectives and Responsibilities in Reducing the Total Amount of Major Pollutants During the 11th Five-Year Plan", which was issued by the China State Council in 2006.

The export data were collected from the China Customs Import and Export Database, which provides a record of all Chinese trade transactions. The initial customs data were aggregated to the 6-digit international Harmonized Commodity Description and Coding System (the Harmonized System, or HS for short) level because there were major adjustments for Chinese HS 8-digit codes, before and after 2002.

The data on emissions intensity for each four-digit industry were collected from China's Environmental Statistics (CES) Database which discloses firm-level emissions monitored by the Ministry of Ecology and Environment of China. The average emissions intensity for each four-digit industry was calculated according to data on firm emissions discharges from that industry.

Based on the firms' location and year, we matched the pollution reduction target data with the China Customs Import and Export Database. Thus, we matched the 6-digit HS

classification with the 4-digit CIC codes using information from the National Bureau of Statistics of China. In this way, we can identify which industry an export product is from.

Another database we used was collected from China's National Bureau of Statistics. It contains all Chinese state-owned enterprises and non-state-owned enterprises whose annual sales are beyond RMB 5 million before 2010 and RMB 20 million after 2010. Before analyzing samples, we excluded the firms' lack of fiscal index such as total assets, total industrial output value, or the net value of fixed assets. Following Feenstra et al. [27] and Brandt et al. [28], we excluded enterprises whose staff is under 8, whose current assets are greater than total assets, whose total fixed assets surpass total assets, whose net value of fixed assets is greater than that of total assets, whose firm identification number is missing, or whose established time is invalid. After that, we matched up with the China Customs Import and Export Database following Yu [29]. Firstly, we matched up with the same year and firm names. In order to maintain more samples, we combined firms with the same zip codes and the last seven digits of their phone numbers. Meanwhile, we deleted firms with no or invalid either phone numbers or zip codes.

After matching the datasets mentioned above, we excluded any firms which miss firm names, export destinations, product categories, unit price, or quantity of each export product. Moreover, firms with single transaction values less than USD 100 or quantities of less than ten, or firms whose exporting destinations were marked as China were also excluded from the sample. In addition, we only kept firms which were operating before and after 2006. Finally, in accordance with Crinò and Ogliari [30], we winsorized all continuous variables at both the top and the bottom 1% in order to remove any outliers. After cleaning the data, the final sample consisted of 89,815 firms, 239 export destinations, and 883 product categories, covering 2001–2010.

### 3.3. Variables

Export product quality (*lnquality*) was the dependent variable. The quality of an export product is the unobserved attribute that influences the way in which consumers perceive the good and their decisions to purchase it, despite high prices [30]. Following Khandelwal et al. [24] and Fan et al. [31], we estimated export product quality using the residual from the following ordinary least squares (OLS) regression:

$$lnx_{fict} + \sigma lnp_{fict} = \varphi_i + \psi_c + \varepsilon_{fict} \tag{2}$$

where $x_{fhct}$ and $p_{fict}$ denote the demand and price of the product in industry $i$ exported by firm $f$ to destination country $c$ in year $t$, respectively. $\varphi_i$ denotes the product fixed effect, and $\psi_c$ denotes the country–year fixed effect. $\varepsilon_{fhct}$ is the error term. Then, the estimated quality is

$$q_{fict} = \frac{\varepsilon_{fict}}{\sigma - 1} \tag{3}$$

where the elasticity of substitution ($\sigma$) was drawn from estimates of [32].

The main independent variables were the $SO_2$ emissions reduction target (*lntarget*1) and the COD emissions target (*lntargets*2) for each province in natural logarithm form. For provinces with a pollution reduction target of zero, we set *lntarget* to zero.

The other independent variable was industrial emissions intensity. The average value of firm $SO_2$ emissions intensity (*lnSO₂*) and COD emissions intensity (*lnCOD*) between 2001 and 2005 for each four-digit industry was used to capture industrial differences. All variables for baseline model are listed in Table 2.

**Table 2.** Definition of variables and summary statistics.

| Variable | Definition | Mean | S.D. | Minimum | Maximum |
|---|---|---|---|---|---|
| *lnquality* | Export product quality calculated by Equations (2)–(4) | 1.2556 | 5.8638 | −19.3053 | 19.2503 |
| *lntarget*1 | $SO_2$ reduction targets for each province | 2.7667 | 0.3837 | 0 | 3.2921 |
| *lntarget*2 | COD reduction targets for each province | 2.6384 | 0.3577 | 0 | 2.7788 |
| *Post* | Dummy variable: 1 for 2006–2010 and 0 otherwise. | 0.5025 | 0.4999 | 0 | 1 |
| $lnSO_2$ | Average $SO_2$ emission intensity for each industry | 1.3377 | 2.1930 | −1.5970 | 5.1330 |
| *lnCOD* | Average COD emission intensity for each industry | 0.7765 | 2.6083 | −3.1281 | 5.7637 |

## 4. Main Results

### 4.1. Baseline Analysis

Table 3 reports the estimated results from Equation (1). When considering the individual emissions plan, the *lntarget*1 $\times$ *Post* $\times$ $lnSO_2$ and *lntarget*2 $\times$ *Post* $\times$ *lnCOD* coefficients in Columns (1)–(2) are negative and pass the significance test at the 1% level. This implies a fall in export product quality in provinces with higher targets following the implementation of a pollution reduction plan, which is sharper the more polluting the industry. Statistically, the export production quality declines 5.74% and 3.31%, respectively, as the pollution reduction target ($SO_2$ and COD) is set 1 unit higher in more polluting industries.

**Table 3.** The baseline modelling results.

| Variables | Dependent Variable: lnquality | | | |
|---|---|---|---|---|
| | **(1)** | **(2)** | **(3)** | **(4)** |
| *lntarget*1 $\times$ *Post* $\times$ $lnSO_2$ | −0.0574 *** (0.0142) | | −0.0437 *** (0.0126) | −0.0425 *** (0.0135) |
| *lntarget*2 $\times$ *Post* $\times$ *lnCOD* | | −0.0331 *** (0.0089) | −0.0142 ** (0.0057) | −0.0153 * (0.0082) |
| Constant | 0.3581 *** (0.0845) | 0.3480 *** (0.0862) | 0.3735 *** (0.0342) | 0.3742 *** (0.0826) |
| Province–year FE | Yes | Yes | No | Yes |
| Industry–year FE | Yes | Yes | No | Yes |
| Province–industry FE | Yes | Yes | No | Yes |
| Firm–destination FE | Yes | Yes | Yes | Yes |
| Observations | 11,794,816 | 11,794,816 | 11,795,228 | 11,794,816 |
| R-squared | 0.2358 | 0.2356 | 0.2270 | 0.2359 |

Notes: Standard errors are corrected for clustering at the province–industry level. *, **, and *** indicate significance at the 10%, 5%, and 1% levels, respectively.

After incorporating the two targeted emissions plans together, our findings showed that the estimated coefficients of *lntarget*1 $\times$ *Post* $\times$ $lnSO_2$ and *lntarget*2 $\times$ *Post* $\times$ *lnCOD* are still negative and significant, with the former being 2.27% higher than the latter in Column (4). Thus, it can be implied that compared with the COD emissions reduction policy, the $SO_2$ emissions reduction policy has greater negative effects on export product quality. This might be because $SO_2$ emissions are the most publicly visible among the targeted pollutants and draw more public attention [33]. Thus, local officials tend to make greater efforts to mitigate $SO_2$ emissions, thus causing sharper declines in export quality.

### 4.2. The Parallel Trend Assumption

Following Hering and Poncet [19], we apply the event study approach to check whether the treatment group and control group have similar pre-treatment trends in their export performance. We choose 2001 as the reference year and explore the dynamic effects of the emissions reduction plans by estimating the following equation:

$$lnquality_{fict} = \beta_0 + \sum_{t=2002}^{t=2010} \beta_t lntarget_{1p} \times Post_t \times lnSO_{2i} + \sum_{t=2002}^{t=2010} \delta_t lntarget_{2p} \times Post_t \times lnCOD_i$$
$$+\mu_{pt} + \gamma_{ht} + \delta_{pi} + \varphi_{fc} + \varepsilon_{fict} \tag{4}$$

where $\{\beta_t; \delta_t\}$ is a series of estimates from 2002 to 2010, and $Post_t$ is the dummy variable indicating the given year $t$. The estimated coefficients are also presented in Appendix A. The coefficients of the triple interactions are insignificant from 2002 to 2005, suggesting a reduction in export quality in polluting industries proportional to the importance of the emissions reduction plans.

### 4.3. Robustness Checks

We conduct a series of robustness checks to confirm our estimation results, including alternative measures and specifications, and controlling for concurrent events, sample selection bias, and the endogeneity problem.

4.3.1. Alternative Measures and Specifications

We estimate a product's quality from both export prices and market share information, following Khandelwal [34]. One concern is that the measurement of export quality may affect our results. Therefore, we follow Hallak and Schott [15] and consider export unit values an alternative measure of export product quality. Moreover, we construct an alternative quality measure by setting the substitution elasticity equal to 5 and 10, which are the lower and higher bounds of substitution elasticity, respectively. Finally, we weight the product-level quality following Crinò and Ogliari [30]. The estimated results are present in Columns (1)–(4) of Table 4. As shown, the coefficients of the triple interaction terms are negative and significant. This suggests that our findings are not driven by alternative measures of the core variables.

**Table 4.** Results of robustness checks I.

| Variables | Dependent Variable: lnquality | | | | |
|---|---|---|---|---|---|
| | **(1)** | **(2)** | **(3)** | **(4)** | **(5)** |
| $lntarget1 \times Post \times lnSO_2$ | −0.0780 *** | −0.0414 *** | −0.0348 *** | −0.0006 *** | −0.1025 *** |
| | (0.0156) | (0.0126) | (0.0109) | (0.0002) | (0.0402) |
| $lntarget2 \times Post \times lnCOD$ | −0.0504 *** | −0.0013 | −0.0181 ** | −0.0002 * | −0.0336 ** |
| | (0.0095) | (0.0108) | (0.0079) | (0.0001) | (0.0152) |
| Constant | 12.5798 *** | 0.5208 *** | 0.6699 *** | 0.7464 *** | 0.8062 *** |
| | (0.0652) | (0.0438) | (0.0677) | (0.0012) | (0.0838) |
| Province–year FE | Yes | Yes | Yes | Yes | Yes |
| Industry–year FE | Yes | Yes | Yes | Yes | Yes |
| Province–industry FE | Yes | Yes | Yes | Yes | Yes |
| Firm–country FE | Yes | Yes | Yes | Yes | Yes |
| Observations | 11,794,816 | 11,794,816 | 11,794,816 | 11,794,816 | 11,794,816 |
| R-squared | 0.3594 | 0.3017 | 0.2272 | 0.2359 | 0.2194 |

Notes: The export product quality in Column (1) is estimated by export unit values; export product quality in Column (2) is estimated by setting the substitution elasticity equal to 5, export product quality in Column (3) is estimated by setting the substitution elasticity equal to 10; export product quality in Column (4) is estimated by the value-weighted average of the product-specific quality estimates; and $SO_2$ and COD in Column (5) are constructed by the average emission intensity between 2001 and 2005 for two-digit industry. Standard errors are corrected for clustering at the province–industry level. *, **, and *** indicate significance at the 10%, 5%, and 1% levels, respectively.

Additionally, we also consider alternative measures of industrial emission intensities. A dummy variable for industrial emission intensity equals 1 if the average $SO_2$/COD discharge intensity between 2001 and 2005 for each two-digit industry is above its mean

value, and 0 otherwise. The emissions data are collected from the Chinese Statistical Yearbook. Consistent estimates are shown in Column (5) of Table 4.

### 4.3.2. Controlling Concurrent Events

We also control for other events during the 11th Five-Year Plan that may simultaneously affect firm performance. Three events stand out: the global financial crisis, the Beijing Olympic Games, and the energy-conservation plan promulgated by the Five-Year-Plan.

To rule out potential effects of the global financial crisis, we delete the samples from 2008 and 2009, and we present our empirical results in Column (1) of Table 5. To investigate the potential effects of the Beijing Olympic Games, we exclude any firms located in either Beijing, Tianjin, Hebei, Liaoning, Shanxi, or Inner Mongolia, as Central and local governments had enacted extreme temporary air quality measures to improve air quality in these areas to ensure environmental quality and fulfil Olympic commitments. The results are shown in Column (2).

**Table 5.** Results of robustness checks II.

| Variables | Dependent Variable: lnquality | | | |
|---|---|---|---|---|
| | **(1)** | **(2)** | **(3)** | **(4)** |
| $lntarget1 \times Post \times lnSO_2$ | −0.0496 *** (0.0120) | −0.0435 *** (0.0147) | −0.0436 *** (0.0133) | −0.0751 *** (0.0118) |
| $lntarget2 \times Post \times lnCOD$ | −0.0241 ** (0.0106) | −0.0174 * (0.0090) | −0.0147 * (0.0079) | −0.0262 ** (0.0126) |
| $lnenergy \times Post \times Dirty$ | | | −1.2738 * (0.7543) | |
| $imr$ | | | | −0.1091 (0.2382) |
| Constant | 0.3066 *** (0.0360) | 0.3322 *** (0.0899) | 2.2458 ** (1.0520) | 0.8012 *** (0.0958) |
| Province–year FE | Yes | Yes | No | Yes |
| Industry–year FE | Yes | Yes | No | Yes |
| Province–industry FE | Yes | Yes | No | Yes |
| Firm–country FE | Yes | Yes | No | Yes |
| Observations | 799,390 | 10,822,124 | 11,794,816 | 11,794,816 |
| R-squared | 0.2562 | 0.2348 | 0.2362 | 0.6813 |

Notes: Results in Column (1) are based on samples excluding financial crisis; Column (2) reports the results excluding the effects of Beijing Olympic Games; Column (3) reports the results controlling the effects of energy-saving plan during the 11th Five-Year Period; and Column (4) reports the second-stage results using Heckman's two-stage method. Standard errors are corrected for clustering at the province–industry level. *, **, and *** indicate significance at the 10%, 5%, and 1% levels, respectively.

Apart from emissions reduction plans, Central and local governments established energy-saving targets during the 11th Five-Year Plan. Thus, this policy is also incorporated into our models. The results are reported in Column (3) of Table 5. They are consistent with the baseline estimates, indicating that our results are robust.

### 4.3.3. Sample Selection Bias

Another concern is sample selection. Our estimations are based on an unbalanced panel, as firms may enter or exit a destination market, and/or add or drop a product from year to year [35]. Therefore, the estimations might be subject to sample selection bias. To correct for possible sample selection bias, we use the two-stage procedure, as proposed by Heckman [36], to run the regression.

Following Li et al. [35], we first estimate a probit selection equation for whether a firm–product–country triplet appears in a given year. The triple interaction terms as well as an exclusive variable marked the "ease of doing business" index from the World Bank's Doing Business data are included. This exclusive variable has a significant effect on a firm's export decisions, but does not affect its export performance [35]. Then, we construct the Inverse Mills Ratio (*imr*) and add it as an additional control variable to the main estimation of export quality equations. The estimated results in Column (4) of Table 5 reveal a significant and negative coefficient. This implies that our main conclusions are still valid.

### 4.3.4. Dealing with Endogeneity

An underlying concern is that some omitted variables which may be associated with pollution reduction targets will lead to endogeneity problems, which our model does not rule out. Following Hering and Poncet [19], we use a ventilation coefficient (the data are collected from European Centre for Medium-Range Weather Forecast), i.e., the average value of the product of wind speed and boundary layer height between 2001 and 2005, as an environmental policy instrument. The results in Column (3) of Table 6 show that in more pollution-intensive industries, stricter emissions reduction target policies are negatively correlated with export product quality. Additionally, we use rainfall and elevation as instrumental variables for environmental policy. The results in Columns (6), (9), and (12) indicate that the triple interactions are negative and significant. Finally, we use a Lagrange multiplier (LM) test and a Wald F test to determine that our instrumental variables are reasonable. Overall, we confirm that our main findings are robust.

### 4.4. Heterogeneity Analysis

In this section, we investigate how the effects differ in terms of firm location and ownership. Columns (1)–(3) in Table 7 show the effects for Eastern, Central, and Western areas. The results show that the coefficient for *lntarget*1 $\times$ *Post* $\times$ *lnSO*$_2$ is $-0.0533$ and it passes the 1% significance tests for Western areas, which are the highest among the three areas. Thus, the negative effects of the SO$_2$ reduction policy are more profound in Western regions. However, the coefficient for *lntarget*2 $\times$ *Post* $\times$ *lnCOD* is negative and significant for Eastern areas while those for Central and Western areas are not significant. The reason might be that the inputs in wastewater treatment facilities are 62.2% higher than those in Central and Western areas, according to the Statistics from China Statistical Yearbook. Inevitably, this leads to increases in firms' production costs, and results in lower export quality.

State-owned and non-state-owned firms may face different pressures to reduce emissions, resulting in different export performance under emissions reduction policies. The coefficients for triple interaction terms are higher in state-owned firms than those in non-state-owned firms. For example, the estimated results presented in Columns (3) and (4) show that the increased SO$_2$ reduction target by 1 unit in more polluting industries leads to a 6.86% decrease in export product quality in state-owned firms. This is 3.35% higher than that in non-state-owned firms. The reason for this might be that state-owned firms shoulder more social responsibility and are inclined to reduce emissions under the pressure of public opinion. Therefore, their environmental performance is superior to that of non-state-owned firms [17,37].

**Table 6.** Two-stage least-squares estimation results.

| Variables | *Policy1* (1) | *Policy2* (2) | *lnquality* (3) | *Policy1* (4) | *Policy2* (5) | *lnquality* (6) | *Policy1* (7) | *Policy2* (8) | *lnquality* (9) | *Policy1* (10) | *Policy2* (11) | *lnquality* (12) |
|---|---|---|---|---|---|---|---|---|---|---|---|---|
| *lnvc × Post × lnSO₂* | 0.3383 *** (0.0181) | −0.0265 (0.0428) | | | | | | | | 0.1061 * 0.0629 | −0.1164 (0.2874) | |
| *lnvc × Post × lnCOD* | 0.0155 (0.0126) | 0.4123 ** (0.0356) | | | | | | | | 0.0485 (0.0063) | 0.2662 (0.3206) | |
| *lnrainfall × Post × lnSO₂* | | | | 0.3318 *** (0.0094) | −0.0252 (0.0388) | | | | | 0.2663 *** (0.0583) | 0.1924 (0.3021) | |
| *lnrainfall × Post × lnCOD* | | | | 0.0196 ** (0.0078) | 0.4106 *** (0.0354) | | | | | −0.0700 (0.0546) | 0.2308 * (0.1284) | |
| *lnelevation × Post × lnSO₂* | | | | | | | −0.3695 *** (0.0828) | −0.4435 *** (0.1178) | | 0.0789 ** (0.0335) | −0.2173 * (0.1116) | |
| *lnelevation × Post × lnCOD* | | | | | | | 0.0739 (0.0746) | 0.7906 *** (0.1015) | | −0.0411 (0.0307) | 0.2824 ** (0.1211) | |
| *lntarget_predict1 × Post × lnSO₂* | | | −0.0892 *** (0.0154) | | | −0.0836 *** (0.0169) | | | −0.0880 *** (0.0186) | | | −0.0851 *** (0.0155) |
| *lntarget_predict2 × Post × lnCOD* | | | −0.0278 * (0.0157) | | | −0.0262 * (0.0160) | | | −0.0233 * (0.0138) | | | −0.0261 ** (0.0128) |
| Province–year FE | Yes | Yes | Yes | Yes | Yes | Yes | Yes | Yes | Yes | Yes | Yes | Yes |
| Industry–year FE | Yes | Yes | Yes | Yes | Yes | Yes | Yes | Yes | Yes | Yes | Yes | Yes |
| Province–industry FE | Yes | Yes | Yes | Yes | Yes | Yes | Yes | Yes | Yes | Yes | Yes | Yes |
| Firm–country FE | Yes | Yes | Yes | Yes | Yes | Yes | Yes | Yes | Yes | | | |
| F-statistic | | 568.25 *** | | | 1382.47 *** | | | 72.10 *** | | | 151.89 *** | |
| Kleibergen–Paap rk LM statistic | | 51.825 *** | | | 49.496 *** | | | 26.009 *** | | | 85.060 *** | |
| Cragg–Donald Wald F statistic | | $5.9 \times 10^6$ *** | | | $5.9 \times 10^6$ *** | | | $2.5 \times 10^6$ *** | | | $2.0 \times 10^6$ *** | |
| Hansen J statistic | | — | | | — | | | — | | | 3.985 [0.4081] | |
| Observations | 11,794,816 | 11,794,816 | 11,794,816 | 11,794,816 | 11,794,816 | 11,794,816 | 11,794,816 | 11,794,816 | 11,794,816 | 11,794,816 | 11,794,816 | 11,794,816 |

Notes: The dependent variables are listed in the first row, where *policy*1 denotes triple interaction terms for *lntarget*1 $\times$ *Post* $\times$ *lnSO₂*, and *policy*2 denotes triple interaction terms for *lntarget*2 $\times$ *Post* $\times$ *lnCOD*. *Lntarget_predict* denotes the predicted variable of pollution reduction plan. The IV models are run by command ivreghdfe in STATA without constant being reported. Standard errors are corrected for clustering at the province–industry level. *, **, and *** indicate significance at the 10%, 5%, and 1% levels, respectively.

**Table 7.** Estimated results of firm heterogeneity.

| Variables | Dependent Variable: lnquality | | | | |
| | Eastern | Central | Western | State | Non-State |
| | (1) | (2) | (3) | (4) | (5) |
|---|---|---|---|---|---|
| $lntarget1 \times Post \times lnSO_2$ | −0.0415 *** | −0.0506 *** | −0.0533 *** | −0.0686 *** | −0.0351 *** |
| | (0.0142) | (0.0173) | (0.0162) | (0.0189) | (0.0136) |
| $lntarget2 \times Post \times lnCOD$ | −0.0165 * | −0.0065 | 0.0056 | −0.0188 * | −0.0143 * |
| | (0.0089) | (0.0059) | (0.0087) | (0.0108) | (0.0082) |
| Constant | 0.3892 *** | 0.1461 *** | 0.1492 ** | 0.0661 | 0.4478 *** |
| | (0.0887) | (0.0518) | (0.0737) | (0.1425) | (0.0696) |
| Province–year FE | Yes | Yes | Yes | Yes | Yes |
| Industry–year FE | Yes | Yes | Yes | Yes | Yes |
| Province–industry FE | Yes | Yes | Yes | Yes | Yes |
| Firm–country FE | Yes | Yes | Yes | Yes | Yes |
| Observations | 11,012,765 | 439,821 | 342,206 | 2,390,569 | 9,404,070 |
| R-squared | 0.2348 | 0.2939 | 0.2012 | 0.0986 | 0.2719 |

Note: Standard errors are corrected for clustering at the province–industry level. *, **, and *** indicate significance at the 10%, 5%, and 1% levels, respectively. F-test of the equality of the coefficients across different columns fail to reject equally at the 5% level in all cases.

## 5. Further Analysis

### 5.1. Potential Mechanism Analysis

As we discussed above, the pollution reduction target policy significantly declines export product quality. By referring to the literature, we find several possible mechanisms through which the emissions reduction policy may affect the quality of the firm's export product in policy and institutional aspects. Our analysis lends itself to both positive and negative mechanisms in the policy aspect. On the one hand, the new environmental policy may influence export quality through compliance cost effects and innovation compensation effects. To meet the requirement of governments' environmental policy, local firms need to control their pollution discharges by updating pollution abatement facilities, or by improving emissions removal rates. This leads to an increase in compliance costs, and impedes the upgrading of export quality [8,38,39]. In addition, the emissions reduction plan also encourages firms to participate in green innovation activities. Accordingly, they can improve their products' export quality [40,41].

Thus, it is widely acknowledged that there is an urgent need for policies that can reduce greenhouse gas emissions [42]. Moreover, for officials who intend to gain trust and be promoted, reducing emissions would boost trust in local government from citizens [43] and they would be more likely to get promoted for those who are able to reduce air pollution [44]. Therefore, local officials' governance motivation is an important factor affecting the implementation effect of emissions reduction plans. Influenced by the promotion incentive, local governments place more emphasis on environmental governance. Thus, the relationship between emissions reduction policies and export quality may be influenced by promotion incentives for local officials.

#### 5.1.1. Policy Mechanisms

To investigate compliance cost effects, we calculate the inputs of pollution abatement facilities by summing up numbers of desulfurization machinery units and wastewater treatment machinery units, and we introduce the logarithm form. We also calculate the firm-level emissions removal rate as the ratio of the total amount of specific pollutant removals over the sum of emissions reductions. Data are collected from the CES Database. Estimated results are reported in Columns (1)–(3) of Table 8. We find statistically significantly positive

coefficients for *lntarget*1 $\times$ *Post* $\times$ *lnSO*$_2$ and *lntarget*2 $\times$ *Post* $\times$ *lnCOD*. This implies that stricter emissions reduction targets improve both the adoption of emissions abatement facilities and emissions removal rates in pollution-intensive industries.

**Table 8.** Results of policy mechanism.

| Variable | lnequipment | lnSO$_2$_remove | lnCOD_remove | lninnovation | lntfp |
|---|---|---|---|---|---|
| | (1) | (2) | (3) | (4) | (5) |
| *lntarget*1 $\times$ *Post* $\times$ *lnSO*$_2$ | 0.0050 ** | 0.0013 *** | | 0.0007 *** | 0.0018 *** |
| | (0.0016) | (0.0004) | | (0.0002) | (0.0005) |
| *lntarget*2 $\times$ *Post* $\times$ *lnCOD* | 0.0021 | | 0.0008 ** | 0.0004 ** | 0.0009 ** |
| | (0.0020) | | (0.0004) | (0.0002) | (0.0004) |
| Constant | 1.1146 *** | 4.2713 *** | 4.2855 *** | 1.2768 *** | 3.4872 |
| | (0.0005) | (0.0001) | (0.0002) | (0.0034) | (0.0052) |
| Province–year FE | Yes | Yes | Yes | Yes | Yes |
| Industry–year FE | Yes | Yes | Yes | Yes | Yes |
| Province–industry FE | Yes | Yes | Yes | Yes | Yes |
| Firm FE | Yes | Yes | Yes | Yes | Yes |
| Observations | 2,419,188 | 3,791,536 | 3,592,861 | 11,794,816 | 6,807,646 |
| R-squared | 0.2938 | 0.6949 | 0.6912 | 0.7194 | 0.7228 |

Note: The dependent variables are listed in the first row. Standard errors are corrected for clustering at the province–industry level. *, **, and *** indicate significance at the 10%, 5%, and 1% levels, respectively.

To test innovation compensation effects, we consider the number of green invention patent applications as a proxy for green innovation (we collected the number and IPC classification of firm-level patent applications from the State Intellectual Property Office (SIPO), and then use the IPC classification number of the green patent in the "International Green Patent Classification List" launched by the World Intellectual Property Organization (WIPO) to identify the number of green patents applied by firms each year) and introduce it in logarithm form. We also use total factor productivity (TFP) as an alternative measure for firm innovation. The estimated results reported in Columns (4)–(5) show that the coefficients for the triple interactions are positive and significant ($p < 5\%$), suggesting that the implementation of emissions reduction targets leads to an increase in firms' green innovation as well as their TFP.

5.1.2. Institutional Mechanisms

China's political reforms have made provincial leaders operate within a well-defined career structure inside the Chinese political hierarchy, wherein an official's age of appointment and their tenure are the most important factors in their behavioral decision making [17,21,45,46]. According to China's Civil Service Retirement System, the age for promotion opportunity is capped at 60 for provincial leaders. Thus, officials close to the age limit confront more promotion pressure and need to demonstrate better environmental performance. Thus, we choose 60 years old as a criterion of approaching retirement and run the estimation separately according to provincial officials' age. Columns (1)–(2) of Table 9 show the results for the provincial governor, while Columns (5)–(6) of Table 9 show the results for the Party secretary. As shown, the coefficients for the triple interactions are greater in samples where provincial leaders are under 60 years old. This implies that the more promotion pressure provincial leaders face, the more detrimental the negative consequences are to export product quality.

**Table 9.** Results of institutional mechanism.

| Variable | Dependent Variable: lnquality | | | | | | | |
|---|---|---|---|---|---|---|---|---|
| | **(1)** | **(2)** | **(3)** | **(4)** | **(5)** | **(6)** | **(7)** | **(8)** |
| $lntarget1 \times Post \times lnSO_2$ | −0.0684 *** | −0.0297 ** | −0.0209 * | −0.0498 *** | −0.0681 *** | −0.0374 *** | −0.0280 *** | −0.0432 *** |
| | (0.0218) | (0.0140) | (0.0125) | (0.0187) | (0.0211) | (0.0138) | (0.0103) | (0.0148) |
| $lntarget2 \times Post \times lnCOD$ | −0.0285 ** | −0.0046 | −0.0072 | −0.0222 * | −0.0290 ** | −0.0142 * | 0.0032 | −0.0182 ** |
| | (0.0130) | (0.0071) | (0.0079) | (0.0116) | (0.0116) | (0.0081) | (0.0075) | (0.0090) |
| Constant | 0.3838 *** | 0.4430 *** | 0.7809 *** | 0.1721 | 0.6416 *** | 0.3100 *** | 0.7028 *** | 0.3429 *** |
| | (0.0599) | (0.0959) | (0.0441) | (0.1197) | (0.0434) | (0.1066) | (0.0319) | (0.0904) |
| Province–year FE | Yes | Yes | Yes | Yes | Yes | Yes | Yes | Yes |
| Industry–year FE | Yes | Yes | Yes | Yes | Yes | Yes | Yes | Yes |
| Province–industry FE | Yes | Yes | Yes | Yes | Yes | Yes | Yes | Yes |
| Firm–country FE | Yes | Yes | Yes | Yes | Yes | Yes | Yes | Yes |
| Observations | 5,716,015 | 6,073,355 | 4,049,314 | 7,743,628 | 3,439,285 | 8,349,772 | 1,057,137 | 10,736,165 |
| R-squared | 0.2443 | 0.2552 | 0.2515 | 0.2332 | 0.2638 | 0.2421 | 0.2568 | 0.2373 |

Note: Columns (1)–(4) show sub-sample results on the provincial governor, while Columns (5)–(8) show sub-sample results on the Party secretary. F-test of the equality of the coefficients across different columns fail to reject equally at the 1% level in all cases. Standard errors are corrected for clustering at the province–industry level. *, **, and *** indicate significance at the 10%, 5%, and 1% levels, respectively.

In addition, the average tenure for provincial governor and Party secretary is roughly 5 years [47,48]. Local officials with longer tenures have greater political incentives to ascend the Party hierarchy. These career concerns incentivize the implementation of Central targets to advance the chances of promotion [49]. In addition, government officials with longer tenures are more familiar with their jobs and have established their own networks of bureaucrats. Therefore, they have more freedom to manage regional GDP numbers [50,51]. Thus, we divide our samples into two subsamples: firms wherein local officials' tenure is 5 years or below, and firms wherein local officials' tenure is more than 5 years. Columns (3)–(4) of Table 9 show the results for the provincial governor, while Columns (7)–(8) of Table 9 show the results for the Party secretary. As shown, the coefficients for the triple interactions are greater in samples where provincial leaders are in office more than 5 years. This suggests that older leaders facing promotion pressures have substantial incentives to engage in emissions reduction plans. This results in lower export product quality.

5.1.3. Summary in Mechanisms

In conclusion, our mechanisms consist of two negative effects and one positive effect which is the innovation compensation effect. It shows that Porter's hypothesis still remains in our research. More importantly, we find that local officials who face promotion pressure are incentivized to raise emissions reduction goals, which may lead to a decrease in export product quality. In addition, increasing emissions abatement costs result in an increase in compliance costs, and it declines the export quality. Under the constraint of the pollution reduction target, the firm's export behavior is affected by both positive incentives and negative limitations. Judging by the baseline and series of robustness analysis, we can corroborate that the negative mechanisms surpass the innovation compensation effect, which leads to a decline in export product quality in general.

*5.2. Mitigating the Negative Effects of Emissions Reduction Plans*

In order to ameliorate the new environmental policy's negative effects, firms usually prefer to concentrate limited resources on core products, or deal with the resulting uncertainty based on new advantages or past experience [16,52,53].

### 5.2.1. Core Product vs. Non-Core Products

Core products are a firm's most efficient and technologically advanced products which dominate the firm's market performance [54,55]. Moreover, core products' market advantage generates knowledge spillovers to non-core products through international production networks [56,57]. A large body of literature indicates that firms will consciously narrow their product range and focus on core products to improve their total factor productivity when facing greater market competition [25,58–60]. Thus, exporting firms can shorten the cognitive distance between their internal product structure and core products to strengthen the technical advantages and spillover effects of their core products. This can reduce their production costs and offset pollution abatement caused by the emissions reduction plans during the 11th Five-Year Period.

Following the approach of Goya and Zahler [61], we define the distance from the core products (*Distance*) as the distance between the top-selling export product in t-1 and each new exported variety (see Appendix B for details). Column (1) of Table 10 presents the results of regressing export product quality on the distance to the core measure defined in Appendix B, the emissions reduction plans, as well as their triple interactions. The estimated results show that the coefficient of *Distance* is negative and significant, indicating that distance from the core product is a significant determinant of export product quality. In addition, the coefficients for the interaction terms between the $SO_2$ reduction plan and distance to the core products (*lntarget*1 × *Post* × *lnSO$_2$* × *Distance*) are still negative and pass the significance test at 1%. This suggests that the negative effects of $SO_2$ reduction plans are more profound for the export quality of non-core products.

**Table 10.** Modelling core products and new advantage.

| Variables | Dependent Variable: lnquality | | | | |
|---|---|---|---|---|---|
| | **(1)** | **(2)** | **(3)** | **(4)** | **(5)** |
| *lntarget*1 × *Post* × *lnSO$_2$* | −0.0484 *** (0.0143) | −0.0271 ** (0.0133) | −0.0463 *** (0.0131) | −0.0423 *** (0.0131) | −0.0636 *** (0.0204) |
| *lntarget*2 × *Post* × *lnCOD* | −0.0155 * (0.0086) | −0.0135 * (0.0072) | −0.0165 ** (0.0082) | −0.0170 ** (0.0078) | −0.0227 ** (0.0104) |
| *Distance* | −0.3015 *** (0.0141) | | | | |
| *Non_Distance* | | 0.3276 *** (0.0174) | | | |
| *First* | | | 0.6736 *** (0.0406) | | |
| *Order* | | | | −0.0001 *** (0.00016) | |
| *lnexperience* | | | | | 0.1490 *** (0.0078) |
| *lntarget*1 × *Post* × *lnSO$_2$* × *Distance* | −0.0114 *** (0.0032) | | | | |
| *lntarget*2 × *Post* × *lnCOD* × *Distance* | −0.0010 (0.0021) | | | | |
| *lntarget*1 × *Post* × *lnSO$_2$* × *Non_Distance* | | 0.0074 *** (0.0014) | | | |
| *lntarget*2 × *Post* × *lnCOD* × *Non_Distance* | | 0.0005 (0.0013) | | | |
| *lntarget*1 × *Post* × *lnSO$_2$* × *First* | | | 0.0321 * (0.0190) | | |

**Table 10.** *Cont.*

| Variables | Dependent Variable: lnquality | | | | |
|---|---|---|---|---|---|
| | **(1)** | **(2)** | **(3)** | **(4)** | **(5)** |
| *lntarget*2 $\times$ *Post* $\times$ *lnCOD* $\times$ *First* | | | 0.0100 *<br>(0.0056) | | |
| *lntarget*1 $\times$ *Post* $\times$ *lnSO$_2$* $\times$ *Order* | | | | $-$0.0001<br>(0.0001) | |
| *lntarget*2 $\times$ *Post* $\times$ *lnCOD* $\times$ *Order* | | | | $-$0.0001<br>(0.0001) | |
| *lntarget*1 $\times$ *Post* $\times$ *lnSO$_2$* $\times$ *lnexperience* | | | | | 0.0050 **<br>(0.0023) |
| *lntarget*2 $\times$ *Post* $\times$ *lnCOD* $\times$ *lnexperience* | | | | | 0.0018 *<br>(0.0011) |
| Constant | 0.2212 ***<br>(0.0836) | 1.5186 ***<br>(0.1345) | 0.3730 ***<br>(0.0824) | 0.3932 ***<br>(0.0818) | $-$0.1347<br>(0.0910) |
| Province–year FE | Yes | Yes | Yes | Yes | Yes |
| Industry–year FE | Yes | Yes | Yes | Yes | Yes |
| Province–industry FE | Yes | Yes | Yes | Yes | Yes |
| Firm–country FE | Yes | Yes | Yes | Yes | Yes |
| Observations | 11,794,816 | 11,794,816 | 11,794,816 | 11,794,816 | 11,794,816 |
| R-squared | 0.2388 | 0.2512 | 0.2361 | 0.2360 | 0.2378 |

Notes: Standard errors are corrected for clustering at the firm–industry–destination level. *, **, and *** indicate significance at the 10%, 5%, and 1% levels, respectively. The models are run by command reghdfe in STATA and singleton observations are dropped automatically.

We also define the non-core distance from the core products (*Non_Distance*) as the distance between the new product and all products except the top-selling one (see Appendix B for details). Column (2) of Table 10 replicates the results for export product quality and survival from Column (1), adding a measure of non-core distance. The coefficient for *Non_distance* is positive and significant. This implies that export quality has improved if the product has a greater distance to the rest of the export basket. Moreover, the coefficient for *lntarget*1 $\times$ *Post* $\times$ *lnSO$_2$* $\times$ *Non_Distance* is positive and significant. This indicates that the negative effects of SO$_2$ reduction plans on export product quality will be mitigated when export categories are closer to their core products. Thus, the view proposed by Goya and Zahler [61] that products further away from an exporter's core competence are less competent is also supported by our analysis.

5.2.2. First-Mover Advantage vs. Late-Mover Advantage

An exporter with a first-mover advantage will innovate and work towards a greater market share by tapping into existing products in new foreign markets [62–64]. Relying on monopolies of technology and/or products, exporting firms can produce excess earnings or extract economic rents, which can mitigate compliance costs caused by environmental regulation [65]. In addition, these exporting firms are confronted with stringent environmental regulations abroad, and they engage in green innovation as well as clean production in order to be compliant with environmental standards. Cumulative foreign experience in solving environmental issues provides a valuable reference for targeted firms [66,67]. Thus, a firm can take more active measures in response to emissions reductions while maintaining export competitiveness.

In line with Hilmersson et al. [68], we measure first-mover advantage by subtracting the year when the firm had its first export value in a given country. The coefficient of the first-mover advantage (*First*) is positive and significant, providing preliminary evidence

that a firm with a first-mover advantage usually has products of higher quality. The triple interaction terms among emissions reduction policy and first-mover advantage (*First*) and its statistical significance remain comparable in Column (3) of Table 10. Combined with the coefficients *lntarget*1 $\times$ *Post* $\times$ *lnSO*$_2$ and *lntarget*2 $\times$ *Post* $\times$ *lnCOD*, we can determine that the negative effects of emissions reduction plans on export product quality are weaker for firms with a first-mover advantage. Specifically, the export product quality in firms with a first-mover advantage is 3.21% higher than those without a first-mover advantage as the pollution reduction target (SO$_2$ and COD) is set 1 unit higher in more polluting industries.

We also define a set of time ranges for the order through which a firm enters into a given market (*Order*). The significant negative coefficient *Order* in Column (4) implies that the later the firms enter foreign markets, the lower the export product quality. However, the coefficients for the triple interaction terms among emissions reduction policy and access order (*order*) are not significant. This indicates that the late-mover advantage has little effect on the relationship between emissions reduction plans and export quality.

Moreover, we also highlight the importance of export experience and present the results in Column (5). Export experience at entry is defined as the duration of a given destination to which a firm already exports. This is in line with Araujo et al. [69]. The coefficients of the triple interaction terms for emissions reduction policy and export experience (*lnexperience*) are positive and significant. Thus, export experience has a profound moderate effect on the correlation between emissions reduction plans and export quality. Therefore, the more experience a firm has in the international marketplace, the weaker the influence of emissions reduction plans on export quality.

## 6. Conclusions and Policy Implications

Along with the acceleration of industrialization in China, environmental pollution, which has received extensive public attention, is even worse than originally thought. Thus, environmental governance is now a prominent problem concerning the national economy and residents' livelihood. Administrative regulation is the most direct and effective method for protecting the environment. At present, the objectives of environmental regulations include both pollution reduction and energy conservation. Thus, it is crucial to examine the trade effects of environmental regulations. Although researchers have tested the effects of environmental regulations on firm exports, research on the relationship between environmental regulations and product-level export quality has been lacking.

We have investigated environmental regulation's effects on export product quality. To control for the potential endogeneity of environmental regulations, we chose the Chinese government's emissions reduction plan proposed in the 11th Five-Year Plan as a quasi-experiment. By using a uniquely detailed dataset comprising Chinese export data at the firm, product, and destination country levels from 2000 to 2010, we find that in more pollution-intensive industries, stricter emissions reduction targets reduce export quality. This finding is robust to a series of robustness checks and the consideration of endogeneity problems, as well as sample selection bias. Furthermore, this negative effect is stronger if the firm is located in Western regions or the firm is state-owned. Our extended results indicate that product switching contributes to resource allocation within firms towards their core products, and mitigates the new environmental policy's effects on export product quality. Additionally, firms can capitalize on their first-mover advantage and past experience to respond to emissions reduction plans while maintaining export quality.

Our findings have important policy implications for realizing environmental protection while simultaneously increasing trade. The relationship between the Central and local governments should be clarified, and a more transparent and practical responsibility mechanism should be established in environmental assessments. Tying emissions quotas to performance in local officials' evaluation systems can promote emissions reduction. Moreover, our mechanism shows that stricter emissions reduction targets increase the adoption of emissions abatement facilities. In this way, local government could assist firms in lowering the costs by raising government subsidies or procurement. In addition, the

Central and local governments should reinforce the effects of innovation offsets through policy support and avoid a one-size-fits-all policy approach in environmental regulation. Further preferential policies supplemented by policies that incentivize innovation should be implemented in under-developed regions and state-owned firms. Finally, governments should formulate favorable policies to guide firms to reallocate their resources into core products by product switching or rebuilding their first-mover advantage.

**Author Contributions:** Conceptualization, X.Z.; methodology, X.Z.; software, X.Z.; formal analysis, X.Z. and H.X.; writing—original draft preparation, X.Z. and H.X.; writing—review and editing, X.Z. and H.X. All authors have read and agreed to the published version of the manuscript.

**Funding:** This research was supported by [the Major Program of National Fund of Philosophy and Social Science of China], grant number [16ZDA038].

**Institutional Review Board Statement:** Not applicable.

**Informed Consent Statement:** Not applicable.

**Data Availability Statement:** The datasets used and/or analyzed during the current study are available from the corresponding author upon reasonable request.

**Conflicts of Interest:** The authors declare no conflicts of interest.

## Appendix A. Regression Results with Lags and Leads

| Variables | Dependent Variable: lnquality |
|---|---|
| $lntarget1 \times lnSO_2 \times Year2002\ Dummy$ | 0.1489 (0.1002) |
| $lntarget1 \times lnSO_2 \times Year2003\ Dummy$ | 0.1455 (0.0930) |
| $lntarget1 \times lnSO_2 \times Year2004\ Dummy$ | 0.1375 (0.0867) |
| $lntarget1 \times lnSO_2 \times Year2005\ Dummy$ | 0.1243 (0.0852) |
| $lntarget1 \times lnSO_2 \times Year2006\ Dummy$ | −0.0523 *** (0.0202) |
| $lntarget1 \times lnSO_2 \times Year2007\ Dummy$ | −0.0458 ** (0.0204) |
| $lntarget1 \times lnSO_2 \times Year2008\ Dummy$ | −0.0413 *** (0.0126) |
| $lntarget1 \times lnSO_2 \times Year2009\ Dummy$ | −0.0377 ** (0.0150) |
| $lntarget1 \times lnSO_2 \times Year2010\ Dummy$ | −0.0427 *** (0.0154) |
| $lntarget1 \times lnCOD \times Year2002\ Dummy$ | 0.0558 (0.0464) |
| $lntarget2 \times lnCOD \times Year2003\ Dummy$ | 0.0615 (0.0461) |
| $lntarget2 \times lnCOD \times Year2004\ Dummy$ | 0.0643 (0.0430) |
| $lntarget2 \times lnCOD \times Year2005\ Dummy$ | 0.0530 (0.0380) |
| $lntarget2 \times lnCOD \times Year2006\ Dummy$ | −0.0151 * (0.0078) |

| Variables | Dependent Variable: Inquality |
|---|---|
| *lntarget2* × *lnCOD* × *Year2007 Dummy* | −0.0095 (0.0100) |
| *lntarget2* × *lnCOD* × *Year2008 Dummy* | −0.0005 (0.0082) |
| *lntarget2* × *lnCOD* × *Year2009 Dummy* | −0.0183 * (0.0103) |
| *lntarget2* × *lnCOD* × *Year2010 Dummy* | −0.0139 (0.0106) |
| Constant | −0.5399 (0.4315) |
| Province–year FE | Yes |
| Industry–year FE | Yes |
| Province–industry FE | Yes |
| Firm–destination FE | Yes |
| Observations | 11,794,816 |
| R-squared | 0.2375 |

Notes: Standard errors are corrected for clustering at the firm–industry–country level. *, **, and *** indicate significance at the 10%, 5%, and 1% levels, respectively.

## Appendix B. Measures of Distance to the Core Products

In line with Goya and Zahler [61], the proximity between products $i$ and $j$ in year t is defined as

$$\phi_{ijt} = min\{Pr[RCA_{it} > 1|RCA_{jt} > 1], \ Pr[RCA_{jt} > 1|RCA_{it} > 1]\} \tag{A1}$$

where $RCA$ is the comparative advantage as defined by Balassa [70].

$$RCA_{it} = \frac{x_{itc}/\sum_i x_{itc}}{\sum_c x_{itc}/\sum_i \sum_c x_{itc}} \tag{A2}$$

where $x_{ict}$ are the exports of product $i$ from country $c$ in year $t$. We then define the distance to the core products as the distance between the product with the largest export value in $t - 1$ and other product variety $i$. Thus,

$$Distance_{fit} = \sum_i s_{fit} \times (1 - \phi_{0it}) \tag{A3}$$

where 0 indicates firm $f$'s core product, with the largest export value in $t - 1$. $S_{fit}$ is the share of product $i$ in the value of exports from $f$ in year $t$.

And the distance to the non-core product is defined as

$$Non\_Distance_{fit} = \sum_i s_{fit} \times (1 - \phi_{it}) - \sum_i s_{fit} \times (1 - \phi_{0it}) \tag{A4}$$

where $\phi_{it}$ is the proximity between products $i$ and other products.

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
