# Peer review of "Emissions Reduction Target Plan and Export Product Quality: Evidence from China’s 11th Five-Year Plan"

_sustainability, doi:10.3390/su16041518_

Round 1
Reviewer 1 Report
Comments and Suggestions for Authors
This manuscript introduces the shock to export product quality in China created by the Chinese Central government’s emissions reduction target policy by the difference-in-difference-in-difference method. The topic is interesting and this study may provide meaningful results to policy maker and English-speaking readers.
However, this manuscript has several limitations that could be improved upon.
1. Limited scope: The study focuses only on the impact of the Chinese Central government's emissions reduction target policy during the 11th Five-Year Plan on export product quality. It does not consider other potential factors that could affect export quality, such as technological advancements or changes in consumer preferences.
2. Lack of control group: The study uses a difference-in-difference-in-difference method, but it does not have a proper control group to compare the effects of the emissions reduction target policy. This makes it difficult to isolate the specific impact of the policy on export product quality.
3. Insufficient analysis of mechanisms: While the study mentions plausible mechanisms driving the results, such as increased emissions abatement costs or diminished green innovation, it does not provide a comprehensive analysis of these mechanisms. Further investigation and analysis of these mechanisms would strengthen the study's findings.
4. Limited generalizability: The study focuses on China's context and does not consider the potential differences in the effects of emissions reduction target policies in other countries or regions. This limits the generalizability of the findings.
To improve these limitations and shortcomings, the following steps could be taken:
1. Expand the scope: Consider incorporating other factors that could affect export product quality, such as technological advancements, changes in consumer preferences, or industry-specific characteristics.
2. Include a control group: Select a suitable control group to compare the effects of the emissions reduction target policy. This would help to isolate the specific impact of the policy on export product quality.
3. Conduct a more in-depth analysis of mechanisms: Explore and analyze the mechanisms driving the observed results in more detail. This could involve conducting additional research or using qualitative methods to gain a deeper understanding of the underlying processes.
4. Consider cross-country or cross-regional comparisons: To enhance the generalizability of the findings, compare the effects of emissions reduction target policies in different countries or regions. This would provide a broader perspective on the relationship between environmental policies and export product quality. By addressing these limitations and shortcomings, future research can provide more robust and comprehensive insights into the impact of emissions reduction target policies on export product quality.
Comments on the Quality of English LanguageThe language in this manuscript can be improved.
Reviewer 2 Report
Comments and Suggestions for Authors
The paper titled "Emission Reduction Targets Plan and Export Product Quality: Evidence from China’s 11th Five-year Plan" explores the impact of the Chinese government's emissions reduction target policy during the 11th Five-Year Plan on export product quality. Below are some comprehensive comments and suggestions on different sections of the paper:
Abstract:
The abstract provides a clear overview of the study. However, it would be beneficial to include specific findings or key results in the abstract to give readers a quick understanding of the study's outcomes.
Introduction:
The introduction effectively sets the context for the study, emphasizing China's environmental challenges and the need for improved export quality.
Consider providing more context on the specific pollutants targeted by the emissions reduction plan (SO2 and COD) to enhance clarity for readers who may not be familiar with these terms.
Clearly state the research questions or hypotheses that the study aims to address.
Literature Review:
The literature review provides a good overview of the existing research on the relationship between environmental policy and export competitiveness.
It would be helpful to include more recent studies in this section, especially those related to China's environmental policies and their economic implications.
Methodology:
The paper mentions the use of the difference-in-difference-in-difference (DDD) model, which is appropriate for assessing the impact of the emissions reduction plan. However, it would be beneficial to provide a brief explanation of how this model works.
Clarify the specific variables used in the analysis, especially those related to emissions reduction targets and export product quality.
Results:
The paper briefly mentions significant negative correlations between the implementation of the pollution reduction target plan and export product quality. Provide more detailed results, including effect sizes and statistical significance levels.
Consider presenting the results in tables or figures to enhance readability.
Discussion:
Expand on the plausible mechanisms mentioned in the paper, such as increases in emissions abatement costs or diminished improvements in green innovation. Provide more in-depth analysis and discussion of these mechanisms.
Discuss the implications of the findings for both policymakers and businesses in China.
Conclusion:
Summarize the key findings concisely and emphasize their significance.
Provide recommendations for future research in this area.
Policy Background Section:
The section on the policy background is comprehensive and provides a good understanding of China's environmental regulations. Consider breaking down the information into subsections for better organization.
Citation:
Include a placeholder for the citation information, as it is currently marked for addition by editorial staff.
Formatting and Language:
Ensure consistent formatting throughout the paper.
Check for grammatical errors and improve sentence structure where necessary.
These suggestions aim to enhance the clarity, completeness, and impact of your research paper. Please feel free to incorporate them based on the specific requirements and guidelines of the journal or platform where you intend to submit the paper.
Methodological Transparency: The paper provides a relatively clear exposition of its methodology, focusing on difference-in-differences models and economic research methods. However, there is room to enhance transparency further. Details on the selection and rationale of control variables in the difference-in-differences model could be more explicit to enhance readers' understanding of the research design.
Robustness Checks: The inclusion of robustness checks, such as alternative measurements and specification tests, is commendable. However, it's advisable to further consider potential endogeneity issues, perhaps by employing more systematic instrumental variable methods.
Data Cleaning and Sample Selection: While the paper mentions data cleaning, specific steps in the data cleaning process are not detailed. Providing more information on data cleaning and sample selection would contribute to the replicability of the study and enhance the credibility of the results.
Diversity in Citations: In the literature review and discussion sections, consider incorporating a more diverse range of literature from related fields to demonstrate a comprehensive understanding of the entire research domain.
Economic Interpretation of Results: In the results discussion, delve deeper into the economic interpretation of the findings and connect them to existing theoretical frameworks in the literature. This would contribute to a more profound theoretical contribution.
Discussion of Results:
The paper effectively discusses the results, especially in terms of compliance cost effects and innovation compensation effects. To strengthen the discussion, consider relating the findings to existing literature and theoretical frameworks.
Policy Implications:
The concluding section provides valuable policy implications. Consider expanding on the practical implementation of these recommendations and potential challenges in policy execution.
Visual Aids:
Consider including visual aids such as graphs or charts to illustrate key findings. This can enhance the accessibility of complex results.
Language and Clarity:
Ensure that the language used is precise and unambiguous. Clearly articulate complex concepts to make the paper accessible to a broader audience.
These comments aim to assist in refining the paper further. Overall, the paper addresses an important issue and provides valuable insights into the relationship between environmental policies and export product quality.
Comments on the Quality of English LanguageMinor editing of English language required
Reviewer 3 Report
Comments and Suggestions for Authors
Controlling and reducing the carbon footprint is one of the most urgent tasks of production, especially in large industrial agglomerations, to improve the environmental situation. The article presents the analysis and the relationship between the carbon footprint and the impact on production, as well as on exports. The presented in-depth analysis allows us to assess the regulatory mechanisms for the carbon footprint of production and export, however, there are several comments on the article, namely:
1. The author of the article should reveal in more detail the system of collecting data on the carbon footprint of enterprises. What mechanism is implemented? Are sensors used or is the calculation based on coefficients and an accurate methodology for each type of production?;
2. The formula for the influence of the product quality factor does not take into account the delivery time, since the duration of deliveries is also a quality criterion. If this information is available, it is proposed to introduce it into the calculation formula;
3. It is necessary to analyze and present graphs of the impact of reducing the carbon footprint of enterprises on the deterioration of quality and decrease in exports.
This work provides a clear picture of the negative impact of carbon footprint restrictions. Conclusion – additional measures are needed on the part of the state to stimulate and support the development of technologies for cleaning and reducing emissions, since the method of restrictions only directly affects the industry.
Round 2
Reviewer 2 Report
Comments and Suggestions for Authors
Agree to accept
Comments on the Quality of English LanguageMinor editing of English language required